# Classification of road traffic injury collision characteristics using text mining analysis: Implications for road injury prevention

**Melita J. Giummarra**[1,2]*, **Ben Beck**[1], **Belinda J. Gabbe**[1,3]

**1** Department of Epidemiology and Preventive Medicine, School of Public Health and Preventive Medicine, Monash University, Melbourne, Victoria, Australia, **2** Caulfield Pain Management and Research Centre, Caulfield Hospital, Caulfield, Victoria, Australia, **3** Health Data Research UK, Swansea University Medical School, Swansea University, Swansea, Wales, United Kingdom

* melita.giummarra@monash.edu

**Data Availability Statement:** The authors do not have approval from the data custodians at the VSTR or the TAC to publish the original data. To access a dataset that is similar to that used in the present study would require a data request to the

## Abstract

Road traffic injuries are a leading cause of morbidity and mortality globally. Understanding circumstances leading to road traffic injury is crucial to improve road safety, and implement countermeasures to reduce the incidence and severity of road trauma. We aimed to characterise crash characteristics of road traffic collisions in Victoria, Australia, and to examine the relationship between crash characteristics and fault attribution. Data were extracted from the Victorian State Trauma Registry for motor vehicle drivers, motorcyclists, pedal cyclists and pedestrians with a no-fault compensation claim, aged > = 16 years and injured 2010–2016. People with intentional injury, serious head injury, no compensation claim/missing injury event description or who died < = 12-months post-injury were excluded, resulting in a sample of 2,486. Text mining of the injury event using QDA Miner and Wordstat was used to classify crash circumstances for each road user group. Crashes in which no other was at fault included circumstances involving lost control or avoiding a hazard, mechanical failure or medical conditions. Collisions in which another was predominantly at fault occurred at intersections with another vehicle entering from an adjacent direction, and head-on collisions. Crashes with higher prevalence of unknown fault included multi-vehicle collisions, pedal cyclists injured in rear-end collisions, and pedestrians hit while crossing the road or navigating slow traffic areas. We discuss several methods to promote road safety and to reduce the incidence and severity of road traffic injuries. Our recommendations take into consideration the incidence and impact of road trauma for different types of road users, and include engineering and infrastructure controls through to interventions targeting or accommodating human behaviour.

## Introduction

By 2030 road traffic injuries are projected to become the fifth leading cause of mortality globally [1]. In Australia, road traffic injuries lead to significant long-term disability and mortality,

Victorian State Trauma Outcomes Registry and Monitoring Group (VSTORM). Instructions available here: https://www.monash.edu/medicine/sphpm/vstorm/data-requests, and data requests require ethics approval before data can be provided. There may be some other limits to data requests. To obtain the linked data from the Transport Accident Commission (TAC) that was used in this study, which was more extensive than the routine linkage between VSTR and TAC, requires a data request via the client research team at the TAC (research@tac.vic.gov.au). To initiate both of these data requests interested parties should first contact the VSTORM project office; contact details are available here: https://www.monash.edu/medicine/sphpm/vstorm/contact. To gain access to the exact same dataset as that used in the present study interested parties should contact the study corresponding author, and follow the same data custodian request processes outlined above.

**Funding:** This project was funded by an Australian Research Council (ARC) Discovery Early Career Research Awards to MJG (DE170100726) and BB (DE180100825), and ARC Future Fellowship to BJG (FT170100048). The Victorian State Trauma Registry is funded by the Transport Accident Commission (TAC) and the Department of Health and Human Services (State Government of Victoria).

**Competing interests:** The authors have declared that no competing interests exist.

especially for people with injuries that require treatment in hospital [2, 3], and have an economic impact exceeding $33 billion AUD per year [4]. All road traffic injury collisions are influenced by a range of factors that are principally attributed to human error, which highlights the need to consider the five safety pillars of the Safe System approach in order to reduce the incidence and impact of road trauma: safe roads, safe people, safe vehicle, safe speeds and post-crash care [5]. To date few studies have sought to characterise the circumstances of road traffic collisions, which presents a key gap in knowledge that impedes our capacity to reduce road trauma.

Many jurisdictions collate road traffic injury data through police, insurance, or government transport agencies. These data can be used to conduct large scale analysis of the incidence of road traffic collisions [6–8]. However, these databases often lack coding of the range of additional individual characteristics and contributory factors involved in each collision. Studies that do generate detailed coding of road traffic injury events typically do so only for specific road user groups [7, 9–12], or focus on specific types of roadways [9, 13, 14]. The aforementioned studies provide important insights to guide injury prevention strategies for specific road users or locations. However, development of injury prevention strategies could be improved further if we understand the events leading to serious injury for all road users who may play a direct role in causing or preventing the road traffic injury event, particularly motor vehicle drivers or motorcycle riders, cyclists and pedestrians.

When seeking to characterise road traffic injury events and to identify avenues for injury prevention, it is helpful to know which party was at fault. Fault can be measured in several ways including recording who was legally responsible (e.g., contributory negligence, intent, knowledge or recklessness of each party) [15]; who has legal liability or which entity must pay compensation for the injuries sustained; and who the injured person blames or feels was responsible for the injury event [16]. The differences between these attributions are important given that people may feel that they are partially to blame even if their actions during the event were not negligent or reckless, or they may recognise that multiple parties were at fault. Fault attributions play a key role in understanding the causal factors contributing to road traffic injury, and are also known to influence outcomes. People who are not responsible, or for whom another party has legal liability to provide injury compensation, have been found to have poorer health, pain and work outcomes after transport injury [16]. While the reasons for this association are not known, it is thought that external attributions of causality and perceptions of injustice for the injury may impede psychological recovery, and negatively influence injury-related beliefs and recovery over time [17]. The benefits of understanding the patterns of fault attributions across different types of road traffic injury collisions may help us to (a) identify where and how injury prevention strategies should be implemented to have the greatest impact on road safety; and (b) identify the types of collisions that may lead to worse injury outcomes, that enable the provision of early targeted interventions to injured road users.

The primary aim of the present study was to classify injury events reported by injured motor vehicle drivers, motorcyclists, pedal cyclists and pedestrians who survive an unintentional road traffic collision using text mining methods. A secondary aim was to examine the associations between road user characteristics, injury characteristics and fault attributions.

## Methods

The study received low risk ethics approval from the Monash University Human Research Ethics Committee (Project number 14283). The study involved analysis of deidentified data. All trauma cases are included in the Victorian State Trauma Registry (VSTR) using an opt out process.

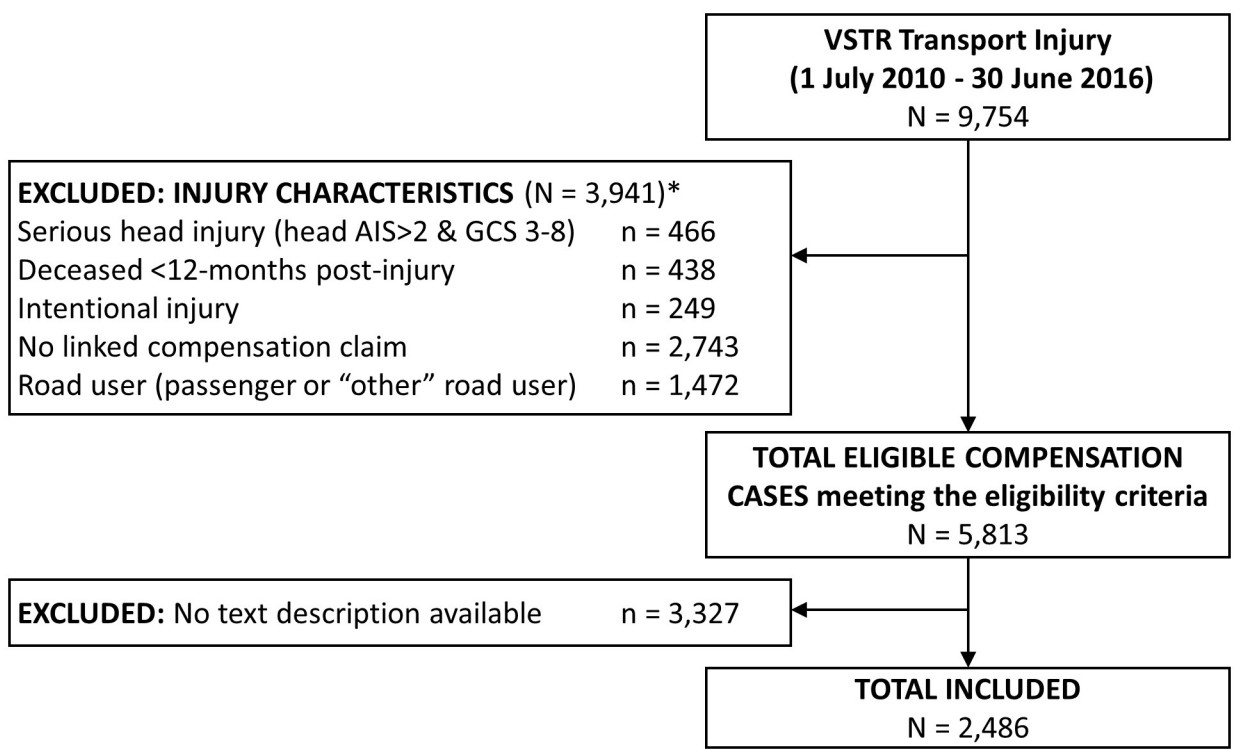

**Fig 1. Participant inclusion chart.** *Notes*: * participants could have met multiple injury-related exclusion criteria.

### Participants

Participants from the VSTR were included if they sustained a road traffic injury as a motor vehicle driver, motorcycle rider, pedal cyclist, or pedestrian between 1 July 2010 and 30 June 2016, were aged 16 years and older at the time of injury, and had an accepted compensation claim with the Transport Accident Commission (TAC). Availability of linked claimant data from the Transport Accident Commission (TAC) was required as the linked claim provided the text description of the injury event, and the claimant and police fault attributions. Motor vehicle and motorcycle passengers were excluded given that the fault data are not likely to represent the passenger's own role in the injury event. People with "other" injury circumstances, predominantly including injuries sustained on a tram, train, mobility scooter or public bus, were excluded due to small numbers. Participants were excluded if their injuries were intentional or if the intent was unknown, if they had a serious head injury (head injury Abbreviated Injury Scale (AIS) score >2 and Glasgow Coma Scale Score ≥3 and ≤8), or if they died within 12 months of injury (Fig 1). People with a serious head injury were excluded as they were considered less likely to be able to report the circumstances of the injury event.

### Setting, data sources and data linkage

The VSTR is a population-based registry that collects information on all patients admitted to one of 138 trauma receiving health services across the state of Victoria who meet major trauma criteria [18]. The inclusion criteria for the VSTR are: (1) death after injury; (2) admission to an intensive care unit (ICU) for > 24 hours and requiring mechanical ventilation for at least part of their ICU stay; (3) Injury Severity Score (ISS) >12; or (4) surgery within 48 hours for intracranial, intrathoracic or intraabdominal injury, or for fixation of pelvic or spinal fractures. The

VSTR includes demographic, pre-injury health, and injury-related characteristics that are collected either from the hospitals or from structured post-discharge telephone interviews. Post-discharge mortality outcomes are determined through linkage with the Victorian Registry of Births, Deaths and Marriages.

The TAC is a government owned organisation that provides financial compensation to people injured in collisions involving at least one motorised vehicle or a vehicle that operates on rails in the State of Victoria, or that involved a vehicle with a Victoria registration. An injured person is entitled to a compensation claim, regardless of fault, to support their healthcare costs if they meet a medical excess, which was $651 during the study period but does not apply to people who are admitted to hospital. People who sustain permanent impairment greater than 10%, as determined by an independent medical examiner, are entitled to lump sum impairment benefits. Additionally, people who are seriously injured and another was partially or completely at fault are entitled to common law compensation.

Claims data from the TAC were accessed from routine data linkage between the VSTR and the TAC using the TAC claim number. Claimant data for people included in the present study were provided to the research team following the routine annual linkage. The TAC provided the claimant's description of the injury event as a special request for this study after they had removed all identifiable nouns (i.e., person and location names).

## Participant demographics

Demographic characteristics from VSTR included age at injury, sex, preferred language, education, work status and occupation pre-injury, and neighbourhood characteristics based on residential postcode at the time of injury. Highest level of education (university, high school, advanced diploma, did not complete high school) was classified in accordance with the Australian Standard Classification of Education (ASCED) [19]. Occupation skill level was classified in accordance with the Australian Standard Classification of Occupations (ASCO) [20]. Occupational skill levels were categorised into six occupation levels: managers and professionals; associate professionals; tradespersons and advanced clerical workers; intermediate sales, clerical, service, production, and transport workers; and elementary sales, clerical, or service workers and labourers.

The Index of Relative Socioeconomic Advantage and Disadvantage (IRSAD) deciles [21] classify neighbourhood socioeconomic position based on national census data on the typical family structure, employment and education level within each postcode region. Victorian ranked IRSAD deciles were summarised into quintiles ranging from one (most disadvantaged) to five (least disadvantaged). The Accessibility/Remoteness Index of Australia (ARIA) (Department of Health and Aged Care, 2001) classifies regions in Australia into five levels of remoteness (major cities, inner regional, outer regional, remote, very remote), which were summarised as major cities versus regional and remote areas due to the small number of remote regions in Victoria.

## Pre-injury health

International Classification of Diseases (10) Australian Modification (ICD-10-AM) diagnosis codes from the hospital coders were used allocate the Charlson Comorbidity Index (CCI) weight [22], and to identify comorbid substance use and mental health conditions. The CCI provides weightings for the severity and number of comorbid conditions, where a weighting of zero indicates no comorbid conditions that increase mortality risk and higher weightings represent greater risk of mortality. The CCI weightings are validated predictors of trauma outcomes following major trauma [23] and orthopaedic injury [24]. Pre-injury substance use and

mental health conditions were identified in accordance with published criteria [25]. Disability level in the week prior to injury was assessed using a five-level rating scale ranging from no disability to severe disability [26].

## Injury characteristics and the injury event description

Injury characteristics included the ISS classified into tertiles, injured body regions based on the maximum AIS body region severity scores, length of hospital stay and discharge destination. Road user group and place of injury were determined from the injury coding in the VSTR. The time of day at which the injury occurred was classified in relation to sunrise and sunset times obtained from Geosciences Australia using criteria that were consistent with previous studies examining injury in relation to daylight hours [14, 27]. We could not classify injury events according to whether they occurred on or off road.

Data from the TAC were used to characterise the injury event including the claimant text description of the injury event, the number of claimants and the number of vehicles involved in the collision, and fault attribution by the claimant and police report. The claim lodgement process includes the question "Was another vehicle at fault in the accident?" with responses of "yes", "no", or "unknown". Fault status was therefore not specific to the potential fault of the injured person's own personal (or their vehicle's) role in the collision. The claimant and police responses were used to categorise participants into the following five groups:

1. *No other at fault*: the claimant and police agree that no other was at fault

2. *Deny another at fault*: the claimant reported no other was at fault, but police reported another was at fault

3. *Another* at fault: the claimant and police agree that another was at fault

4. *Claim another at fault*: the claimant reported another was at fault, but police reported no other was at fault

5. *Unknown*: neither the claimant nor police know whether another was at fault

## Data analysis

The data were processed and analysed using QDA Miner version 5, Wordstat 7.1.22 and Stata Version 15.0. The association between participant demographic, health and injury-related characteristics and fault attribution were examined for each road user group using Chi Square tests and Kruskal-Wallis test in order to provide a descriptive overview of characteristics associated with fault attributions.

The text descriptions of injury events were analysed in Wordstat. The text was first corrected of grammatical errors, and processed using lemmatization and categorisation dictionaries that were iteratively developed from examining the keywords and phrases that appeared in the corpus. Cluster analyses were used to help identify the terms used to describe each type of injury event, which were exported to Stata for analysis. A detailed explanation of the text analysis methods is available in the S4 File; however, in brief, the text analysis followed the six steps outlined below:

1. Removal of punctuation and symbols, and **correction of spelling errors**.

2. Applying **exclusion list** (S1 File) for cluster and keyword co-occurrence analyses.

3. Viewing keywords in context and applying an adaptation of the English **lemmatization** dictionary (S2 File) developed by Provalis Ltd to consolidate terms to be processed.

4. Developing a **categorisation dictionary** (S3 File) based on the review of keywords and phrases so that semantically similar concepts could be analysed as a single category of terms.

5. Undertaking multiple iterations of **exploratory cluster analyses** for each road user group to review the commonly co-occurring keywords and category terms. Cluster terms were then reviewed in context to identify the combinations of keywords and category terms that are used to describe similar types of injury events.

6. Extracting keyword occurrences to Stata for classification of injury events, and identification of the relevant VicRoads Definitions for Classifying Accidents [DCA; 28] categories.

The DCA is used to classify collision and near-collision events from the perspective of the driver of a vehicle, and the key categories for this study included: pedestrian impacts (DCA 100–108); turning vehicles from an adjacent direction (DCA 110–118) or from opposing direction (DCA 121–125); head on collision (DCA 120); collisions in the same direction (i.e., rear-end collisions; DCA 130–137); manoeuvring (e.g., u turns, leaving or entering a car park; DCA 140–148); overtaking or changing lanes (DCA 150–154); collision with an animal (DCA 167); events in which the vehicle lost control on path (DCA 160–167) or off path (DCA 170–175) on straight, or on a curve (DCA 180–184); and other miscellaneous events involving falls from a vehicle, being struck by a load falling from a vehicle, hitting a train/tram/railway infrastructure, or being hit by a runaway parked car (DCA 190–194). Injury events could be classified into more than one scenario. Injury events were analysed separately for each road user group, and were summarised using frequencies and percentages.

## Results

### Cohort overview

There was a total of 9,754 cases admitted to hospital following road traffic injuries who met major trauma criteria for inclusion in VSTR between 1 July 2010 and 30 June 2016. Of those cases, 3,941 were excluded from this study due to the injury and compensation claim eligibility criteria (Fig 1). Of the 5,813 motor vehicle drivers, motorcyclist, pedal cyclists and pedestrians with a compensation claim who met the inclusion criteria, only 2,486 could be included in the study as their claim included a description of the injury event. A higher proportion of cases whose claim did not include a description of the injury event spoke English as their preferred language, lived in neighbourhoods with greater disadvantage, were working pre-injury, had a pre-existing substance use or mental health condition, sustained more severe injuries, were involved in collisions with fewer vehicles, were injured in the evening hours, and when another vehicle was at fault (Table 1). The text descriptions were only available for people with injuries between 2013 to 2016, except for two cases from 2011–12.

The cohort had a median age of 43 years (Q1-Q3: 28–60), and were predominantly male (n = 1782, 71.7%). Motorcyclists had the youngest median age of all injured road users (Median (Med) = 38, Q1-3: 27–50) compared with motor vehicle drivers (Med = 44, Q1-Q3: 28–66) and pedal cyclists (Med = 44, Q1-3: 34–74), and pedestrians were the oldest injured road users (Med = 54, Q1-3: 30–74), p<0.001. A higher proportion of motorcyclists were male (95.2%) compared with pedal cyclists (79%), motor vehicle drivers (60%) and pedestrians (53%), p<0.001. Eighty eight percent of injured pedal cyclist and pedestrians lived in metropolitan areas, compared with 62.8% (n = 712) of motor vehicle drivers and 76.9% (n = 598) of motorcycle rider, p < 0.001. While 69% of pedal cyclists were living in neighbourhoods with the highest two quintiles of socioeconomic position, the other road user groups were relatively

**Table 1. Characteristics of the participants included compared with people who did not have a description of their injury event, and could not be included, n(%), N = 5,813.**

| | | Included (n = 2486) | Not included (n = 3327) | p |
|---|---|---|---|---|
| Sex | Male | 1782 (71.7) | 2464 (74.1) | 0.043 |
| | Female | 704 (28.3) | 863 (25.9) | |
| Age group (years) | 15 to 24 | 435 (17.5) | 654 (19.7) | 0.003 |
| | 25 to 34 | 463 (18.6) | 626 (18.8) | |
| | 35 to 44 | 429 (17.3) | 590 (17.7) | |
| | 45 to 54 | 366 (14.7) | 506 (15.2) | |
| | 55 to 64 | 301 (12.1) | 414 (12.4) | |
| | 65 to 74 | 226 (9.1) | 284 (8.5) | |
| | 75+ | 266 (10.7) | 253 (7.6) | |
| Education level [a] | University | 453 (18.7) | 562 (17.9) | 0.54 |
| | Completed high school | 314 (13.0) | 429 (13.7) | |
| | Advanced diploma | 775 (32.1) | 971 (30.9) | |
| | Did not complete high school | 876 (36.2) | 1178 (37.5) | |
| Region of residence [b] | Regional and remote areas | 954 (29.2) | 667 (27.2) | 0.098 |
| | Major cities | 2318 (70.8) | 1788 (72.8) | |
| IRSAD, quintiles [b] | 1, most disadvantaged | 440 (17.9) | 624 (19.1) | 0.009 |
| | 2 | 406 (16.5) | 641 (19.6) | |
| | 3 | 507 (20.7) | 610 (18.6) | |
| | 4 | 508 (20.7) | 618 (18.9) | |
| | 5, least disadvantaged | 594 (24.2) | 779 (23.8) | |
| English preferred language | Yes | 1753 (70.5) | 2745 (82.5) | <0.001 |
| | No | 733 (29.5) | 582 (17.5) | |
| Working pre-injury [c] | No | 725 (31.2) | 802 (25.9) | <0.001 |
| | Yes | 1597 (68.8) | 2293 (74.1) | |
| Occupation group [d] | Managers/ professionals | 413 (28.8) | 505 (24.3) | <0.001 |
| | Associate professionals | 126 (8.8) | 208 (10.0) | |
| | Trade/ advanced clerical | 467 (32.6) | 611 (29.5) | |
| | Intermediate | 224 (15.6) | 431 (20.8) | |
| | Elementary/labourers | 203 (14.2) | 319 (15.4) | |
| CCI weighting | 0 | 1884 (75.8) | 2506 (75.3) | <0.001 |
| | 1 | 427 (17.2) | 669 (20.1) | |
| | >1 | 175 (7.0) | 152 (4.6) | |
| Pre-injury substance use condition [e] | No | 2297 (92.8) | 2977 (90.4) | 0.001 |
| | Yes | 177 (7.2) | 315 (9.6) | |
| Pre-injury mental health condition [e] | No | 2292 (92.6) | 2968 (90.2) | <0.001 |
| | Yes | 182 (7.4) | 324 (9.8) | |
| Pre-injury disability [f] | None | 1778 (83.7) | 2459 (87.3) | 0.001 |
| | Mild | 232 (10.9) | 226 (8.0) | |
| | Moderate to severe | 113 (5.3) | 132 (4.7) | |
| Injury season | Summer | 683 (27.5) | 786 (23.6) | <0.001 |
| | Autumn | 705 (28.4) | 890 (26.8) | |
| | Winter | 478 (19.2) | 829 (24.9) | |
| | Spring | 620 (24.9) | 822 (24.7) | |
| Injury Time of day [g] | 1hr before sunrise | 65 (2.8) | 106 (3.5) | <0.001 |
| | 1hr after sunrise | 97 (4.2) | 127 (4.2) | |
| | daylight | 1470 (63.0) | 1756 (58.3) | |
| | 1hr before sunset | 127 (5.4) | 149 (4.9) | |
| | 1hr after sunset | 129 (5.5) | 161 (5.3) | |
| | dark (evening/early) | 444 (19.0) | 712 (23.6) | |

*(Continued)*

**Table 1.** (Continued)

| | | Included (n = 2486) | Not included (n = 3327) | p |
|---|---|---|---|---|
| Place of injury | Road, street or highway | 2282 (91.8) | 3046 (91.6) | 0.74 |
| | Other | 204 (8.2) | 281 (8.4) | |
| Road user group | Motor vehicle driver | 1150 (46.3) | 1585 (47.6) | 0.041 |
| | Motorcyclist | 786 (31.6) | 941 (28.3) | |
| | Pedal cyclist | 196 (7.9) | 281 (8.4) | |
| | Pedestrian | 354 (14.2) | 520 (15.6) | |
| Length of hospital stay | < = 2 days | 181 (7.3) | 238 (7.2) | 0.032 |
| | 3–6 days | 1108 (44.6) | 1407 (42.3) | |
| | 7–13 days | 824 (33.1) | 1088 (32.7) | |
| | > = 14 days | 373 (15.0) | 594 (17.9) | |
| Injury severity (ISS), tertiles | < = 10 | 964 (38.8) | 1120 (33.7) | <0.001 |
| | 11–17 | 891 (35.8) | 1268 (38.1) | |
| | > = 18 | 631 (25.4) | 939 (28.2) | |
| Nature of injury | Isolated head injury | 34 (1.4) | 49 (1.5) | 0.003 |
| | Head/other | 259 (10.4) | 413 (12.4) | |
| | Spinal Cord Injury | 25 (1.0) | 43 (1.3) | |
| | Orthopaedic injury only | 841 (33.8) | 957 (28.8) | |
| | Chest/abdominal injuries only | 81 (3.3) | 113 (3.4) | |
| | Chest/abdominal/other | 759 (30.5) | 1069 (32.1) | |
| | Other/multi-trauma | 487 (19.6) | 683 (20.5) | |
| Discharge destination | Home | 1245 (50.1) | 1873 (56.3) | <0.001 |
| | Other | 1241 (49.9) | 1454 (43.7) | |
| Fault group | No other at fault | 1107 (44.5) | 1346 (40.5) | <0.001 |
| | Deny another at fault | 148 (6.0) | 59 (1.8) | |
| | Another at fault | 399 (16.0) | 904 (27.2) | |
| | Claim another at fault | 224 (9.0) | 146 (4.4) | |
| | Unknown | 608 (24.5) | 872 (26.2) | |
| Number of claimants | Single claimant | 1927 (77.5) | 2009 (76.0) | 0.20 |
| | Multiple claimants | 559 (22.5) | 634 (24.0) | |
| Number of vehicles | 1 | 892 (35.9) | 1272 (38.2) | <0.001 |
| | 2 | 718 (28.9) | 1083 (32.6) | |
| | 3 | 415 (16.7) | 215 (6.5) | |
| | 4 of more | 461 (18.5) | 757 (22.8) | |

*Abbreviations*: *Abbreviations*: CCI = Charlson Comorbidity Index; hr = hour; IRSAD = Index of Relative Social Advantage and Disadvantage; ISS = Injury Severity Score.

[a] Missing n = 255;

[b] Missing n = 86;

[c] Missing n = 396;

[d] Missing n = 70;

[e] Missing n = 47;

[f] Missing n = 873;

[g] Missing n = 468.

evenly distributed across IRSAD quintiles, p<0.001. Most cases were the sole claimant from their injury event (n = 1927, 77.5%), even though 64.1% of all collisions involved two or more vehicles. In particular, 40% of motor vehicle driver collisions had multiple claimants compared with only 6–9% of the other injured road user groups, p<0.001.

The characteristics associated with fault attribution groups for motor vehicle drivers (n = 1150), motorcyclists (n = 786), pedal cyclists (n = 196) and pedestrians (n = 354) are reported in Tables 2–5, respectively. For **motor vehicle drivers** the only characteristics that differed across fault attribution groups was that a larger proportion of people who were working pre-injury were injured when another was at fault, or claimed to be at fault. Even though the majority of drivers did not have a substance use (91.9%) or mental health condition (92.3%), it was notable that a higher proportion of people injured in collisions where no other was at fault did have a substance use condition. A higher proportion of drivers who were the sole claimant or who were injured in single vehicle collision reported that no other vehicle was at fault, denied that another was at fault, or did not know if another was at fault. A higher proportion of **motorcyclists** who were injured at the fault of another were working pre-injury and were a median of 7–11 years older than the other fault groups. A higher proportion of motorcyclists who had a pre-existing substance use condition were injured in collisions in which no other was at fault, and a larger proportion of motorcyclists with unknown fault had preinjury disability. Most motorcyclists were the sole claimant from their injury event; however, a higher proportion of motorcyclists with no other at fault, or who denied another was at fault, were injured in collisions with a single vehicle compared with the other fault groups.

Most **pedal cyclists** (61.7%) and **pedestrians** (49.4%) reported unknown fault. No demographic, health and injury characteristics varied across fault groups for pedal cyclists. Pedestrians who reported unknown fault were a median of 20 years older and were predominantly unemployed pre-injury compared with pedestrians who reported whether another party was at fault or not. A higher proportion of pedestrians who reported that no other was at fault, or denied that another was at fault had a CCI weighted condition pre-injury.

## Injury events

The total text corpus for injury event descriptions contained a total of 40,057 words (Table 6). Each case contained a median of 11 words (Q1-3: 6–18). Across all road user groups 229 (9.2%) cases had no recollection of the injury event (Fig 2). Sixty eight percent of motorcyclists and 84% of motor vehicle drivers with no recollection of the injury event were injured in collisions where no other was at fault, or in which they denied another was at fault. On the contrary, 48% and 57% of pedestrians and pedal cyclists, respectively, were injured in collisions where the fault of another vehicle was unknown.

Most injury events occurred during daylight hours (Fig 3). There were no apparent differences in the number of injuries occurring throughout the week for motor vehicle drivers; however, a larger proportion of motorcyclists were injured on the weekend than during the week, a larger proportion of pedal cyclists were injured between Tuesday and Thursday, and pedestrian injuries peaked on Fridays.

## Motor vehicle driver injury event classifications

There were 34 motor vehicle driver collision classifications. The number of vehicles and fault attribution groups involved in each type of collision varied, as described below and shown in Fig 4 for the most common collision types. The majority of driver collisions in which **no other was at fault**, or the driver **denied that another was at fault**, involved losing control of the vehicle and/or veering off the road, including losing control when driving over or onto earth-matter (e.g., dirt, gravel or stones) or plant-matter (e.g., branches on the road) on or by the side of the road. All but one collision type occurred on a road, street or highway according to the place of injury classification by VSTR. Other circumstances with no other at fault involved poor weather, road conditions, or limited visibility; having a mechanical failure; hitting a tram

**Table 2. Characteristics of motor vehicle drivers stratified by fault attribution, n(%), N = 1,150.**

| | | No other at fault | | Another at fault | | | |
|---|---|---|---|---|---|---|---|
| | | No other at fault (n = 688) | Deny another at fault (n = 31) | Another at fault (n = 178) | Claim another at fault (n = 76) | Unknown Fault (n = 177) | p-value |
| Sex | Male | 426 (61.9) | 20 (64.5) | 91 (51.1) | 45 (59.2) | 109 (61.6) | 0.12 |
| | Female | 262 (38.1) | 11 (35.5) | 87 (48.9) | 31 (40.8) | 68 (38.4) | |
| Age group (years) | Median [Q1,Q3] | 44 [27, 67] | 46 [28, 63] | 49 [35, 63] | 34 [25, 61] | 44 [29, 67] | 0.21 |
| Age group (years) | 15–24 | 131 (19.0) | 5 (16.1) | 20 (11.2) | 17 (22.4) | 33 (18.6) | 0.002 |
| | 25–34 | 114 (16.6) | 5 (16.1) | 22 (12.4) | 22 (28.9) | 25 (14.1) | |
| | 35–44 | 106 (15.4) | <5 | 35 (19.7) | 7 (9.2) | 34 (19.2) | |
| | 45–54 | 77 (11.2) | 7 (22.6) | 29 (16.3) | 6 (7.9) | 20 (11.3) | |
| | 55–64 | 69 (10.0) | <5 | 32 (18.0) | 10 (13.2) | 16 (9.0) | |
| | 65–74 | 76 (11.0) | <5 | 25 (14.0) | 6 (7.9) | 16 (9.0) | |
| | 75+ | 115 (16.7) | 4 (12.9) | 15 (8.4) | 8 (10.5) | 33 (18.6) | |
| Education level | University degree | 95 (14.2) | 5 (17.2) | 40 (23.4) | 15 (20.8) | 23 (13.5) | 0.044 |
| | Completed high school | 109 (16.3) | 5 (17.2) | 27 (15.8) | 6 (8.3) | 18 (10.6) | |
| | Advanced diploma | 190 (28.5) | 11 (37.9) | 51 (29.8) | 19 (26.4) | 56 (32.9) | |
| | Did not complete high school | 273 (40.9) | 8 (27.6) | 53 (31.0) | 32 (44.4) | 73 (42.9) | |
| Region of residence IRSAD, quintiles | Regional and remote | 253 (37.2) | 14 (46.7) | 54 (31.2) | 23 (31.1) | 78 (44.3) | 0.066 |
| | Major cities | 428 (62.8) | 16 (53.3) | 119 (68.8) | 51 (68.9) | 98 (55.7) | |
| | 1, most disadvantaged | 144 (21.1) | 7 (23.3) | 42 (24.3) | 12 (16.2) | 24 (13.6) | 0.62 |
| | 2 | 128 (18.8) | 4 (13.3) | 36 (20.8) | 13 (17.6) | 35 (19.9) | |
| | 3 | 143 (21.0) | 9 (30.0) | 36 (20.8) | 17 (23.0) | 39 (22.2) | |
| | 4 | 128 (18.8) | 5 (16.7) | 28 (16.2) | 11 (14.9) | 36 (20.5) | |
| | 5, least disadvantaged | 138 (20.3) | 5 (16.7) | 31 (17.9) | 21 (28.4) | 42 (23.9) | |
| Preferred language English | Yes | 482 (70.1) | 28 (90.3) | 125 (70.2) | 53 (69.7) | 132 (74.6) | 0.13 |
| | No | 206 (29.9) | <5 | 53 (29.8) | 23 (30.3) | 45 (25.4) | |
| Working prior | No | 267 (42.0) | 13 (41.9) | 50 (28.7) | 23 (31.9) | 77 (47.5) | 0.003 |
| | Yes | 368 (58.0) | 18 (58.1) | 124 (71.3) | 49 (68.1) | 85 (52.5) | |
| Occupation skill level | Professionals/Associate professionals | 103 (31.8) | 5 (27.8) | 44 (39.3) | 19 (41.3) | 29 (40.8) | 0.32 |
| | Trade/advanced clerical/ elementary | 221 (68.2) | 13 (72.2) | 68 (60.7) | 27 (58.7) | 42 (59.2) | |
| CCI weighted condition | None | 496 (72.1) | 20 (64.5) | 132 (74.2) | 60 (78.9) | 131 (74.0) | 0.56 |
| | 1 or more | 192 (27.9) | 11 (35.5) | 46 (25.8) | 16 (21.1) | 46 (26.0) | |
| Pre-injury substance use condition | No | 615 (89.9) | 31 (100.0) | 174 (97.8) | 70 (92.1) | 160 (92.5) | 0.005 |
| | Yes | 69 (10.1) | <5 | <5 | 6 (7.9) | 13 (7.5) | |
| Pre-injury mental health condition | No | 625 (91.4) | 30 (96.8) | 171 (96.1) | 71 (93.4) | 157 (90.8) | 0.20 |
| | Yes | 59 (8.6) | <5 | 7 (3.9) | 5 (6.6) | 16 (9.2) | |
| Pre-injury disability | None | 458 (79.5) | 23 (79.3) | 142 (86.6) | 49 (80.3) | 110 (75.3) | 0.16 |
| | Mild to severe | 118 (20.5) | 6 (20.7) | 22 (13.4) | 12 (19.7) | 36 (24.7) | |
| Place of injury | Road, street or highway | 675 (98.1) | 29 (93.5) | 178 (100.0) | 74 (97.4) | 171 (96.6) | 0.069 |
| | Other | 13 (1.9) | <5 | 0 | <5 | 6 (3.4) | |
| Injury severity (ISS), tertiles | < = 10 | 258 (37.5) | 12 (38.7) | 66 (37.1) | 37 (48.7) | 62 (35.0) | 0.17 |
| | 11–17 | 247 (35.9) | 15 (48.4) | 65 (36.5) | 28 (36.8) | 61 (34.5) | |
| | > = 18 | 183 (26.6) | 4 (12.9) | 47 (26.4) | 11 (14.5) | 54 (30.5) | |

(*Continued*)

**Table 2.** (Continued)

| | | No other at fault | | Another at fault | | | |
|---|---|---|---|---|---|---|---|
| | | No other at fault (n = 688) | Deny another at fault (n = 31) | Another at fault (n = 178) | Claim another at fault (n = 76) | Unknown Fault (n = 177) | p-value |
| Nature of injury | Orthopaedic injuries | 209 (30.4) | 11 (35.5) | 50 (28.1) | 26 (34.2) | 44 (24.9) | 0.64 |
| | Head or SCI | 77 (11.2) | 1 (3.2) | 18 (10.1) | 6 (7.9) | 22 (12.4) | |
| | Chest, abdominal or multi trauma injuries | 402 (58.4) | 19 (61.3) | 110 (61.8) | 44 (57.9) | 111 (62.7) | |
| Time of day + | 1hr after sunrise | 22 (3.4) | <5 | 7 (4.2) | <5 | 5 (3.2) | 0.98 |
| | 1hr before sunset | 37 (5.6) | <5 | 10 (6.0) | 5 (6.6) | <5 | 0.35 |
| | 1hr after sunset | 28 (4.3) | <5 | 8 (4.8) | <5 | 5 (3.2) | 0.88 |
| Number of claimants | Single claimant | 443 (64.4) | 23 (74.2) | 70 (39.3) | 43 (56.6) | 107 (60.5) | <0.001 |
| | Multiple claimants | 245 (35.6) | 8 (25.8) | 108 (60.7) | 33 (43.4) | 70 (39.5) | |
| Number of vehicles | One | 327 (47.5) | 24 (77.4) | 13 (7.3) | 19 (25.0) | 46 (26.0) | <0.001 |
| | Two or more | 361 (52.5) | 7 (22.6) | 165 (92.7) | 57 (75.0) | 131 (74.0) | |

*Notes*: All comparisons tested with Chi Square test, except for age (Kruskal-Wallis);

+ Time of day coded as dummy variables comparing the hour before or after sunrise and sunset with all other times of the day.

*Abbreviations*: CCI = Charlson Comorbidity Index; IRSAD = Index of Relative Socioeconomic and Disadvantage; ISS = Injury Severity Score; Q = quartile.

or bus; or hitting an animal or losing control while trying to avoid an animal that had entered the road. Losing control was reported by 492 (42.8%) drivers, 353 (71.8%) of whom reported that no other was at fault and 253 (51.6%) of whom were injured in a single-vehicle collision. Mechanical failures for motor vehicle drivers primarily included having a flat or blown tyre, failed brakes, or when the driver stated that their foot was stuck on the accelerator. Driver injury events in which there was predominantly **another at fault**, or a **claim that another was at fault**, involved another vehicle entering the road or the driver's lane; being hit while the driver's vehicle was stationary or slowing in traffic; being in a rear-end or t-bone collision; or losing control when trying to avoid an obstacle or another vehicle. Driver collision classifications that had a higher proportion of cases (>10% of the collision scenarios) with **unknown fault** included scenarios in which the driver was hit while stationary; the driver hit a pole; the collision involved hitting or being hit by another vehicle, or a t-bone or rear-end collision with another vehicle. In particular, collisions involving two or more vehicles had a high proportion of cases with unknown fault (35.3% of all cases involving two or more vehicles).

A large number of cases had a collision with a tree (n = 166), pole (n = 55) or other type of barrier (e.g., fence, house, wall or safety railing; n = 20). Sixty two cases involved a collision with a heavy vehicle, and the vehicle rolled in 91 "loss of control" injury events. Collision classifications with a frequency of <10 cases that are not depicted in Fig 4 included: cases who reported being unable to stop in time to avoid an obstacle; loss of control while towing a trailer or caravan, or travelling around a corner or bend; collision with a rock on or at the side of the road; collision while reversing; being hit by another vehicle and going into an embankment; and disclosure that the driver was under the influence of illicit substances or alcohol.

### Motorcyclist injury event classifications

There were 35 motorcyclist collision classifications, of which 31 are summarised in Fig 5. Crash classifications where there was predominantly **no other at fault**, or where motorcyclists **denied that another was at fault,** included scenarios in which the motorcyclist described that they fell off or lost control while avoiding an obstacle, or attempted to avoid an obstacle; lost

**Table 3. Characteristics of motorcyclists stratified by fault attribution, n(%), N = 786.**

| | | No other at fault | | Another at fault | | | |
|---|---|---|---|---|---|---|---|
| | | No other at fault (n = 351) | Deny another at fault (n = 88) | Another at fault (n = 153) | Claim another at fault (n = 59) | Unknown Fault (n = 135) | p-value |
| Sex | Male | 330 (94.0) | 85 (96.6) | 146 (95.4) | 57 (96.6) | 130 (96.3) | 0.72 |
| | Female | 21 (6.0) | 3 (3.4) | 7 (4.6) | 2 (3.4) | 5 (3.7) | |
| Age group (years) | Median [Q1, Q3] | 35 [26, 47] | 43 [29, 52] | 42 [31, 52] | 32 [27, 51] | 38 [26, 55] | 0.007 |
| Education level | University degree | 45 (12.9) | 15 (17.2) | 36 (23.7) | 4 (6.8) | 17 (12.9) | 0.005 |
| | Completed high school | 38 (10.9) | 11 (12.6) | 13 (8.6) | 10 (16.9) | 12 (9.1) | |
| | Advanced diploma | 158 (45.1) | 39 (44.8) | 67 (44.1) | 19 (32.2) | 50 (37.9) | |
| | Did not complete high school | 109 (31.1) | 22 (25.3) | 36 (23.7) | 26 (44.1) | 53 (40.2) | |
| Region of residence | Regional and remote | 89 (25.5) | 31 (35.2) | 24 (15.7) | 7 (11.9) | 32 (24.2) | 0.002 |
| | Major cities | 260 (74.5) | 57 (64.8) | 129 (84.3) | 52 (88.1) | 100 (75.8) | |
| IRSAD, quintiles | 1, most disadvantaged | 58 (16.6) | 11 (12.5) | 18 (11.8) | 9 (15.3) | 28 (21.2) | 0.13 |
| | 2 | 68 (19.5) | 16 (18.2) | 19 (12.4) | 9 (15.3) | 21 (15.9) | |
| | 3 | 76 (21.8) | 21 (23.9) | 34 (22.2) | 13 (22.0) | 31 (23.5) | |
| | 4 | 77 (22.1) | 12 (13.6) | 36 (23.5) | 15 (25.4) | 31 (23.5) | |
| | 5, least disadvantaged | 70 (20.1) | 28 (31.8) | 46 (30.1) | 13 (22.0) | 21 (15.9) | |
| Preferred language English | Yes | 269 (76.6) | 70 (79.5) | 106 (69.3) | 45 (76.3) | 99 (73.3) | 0.36 |
| | No | 82 (23.4) | 18 (20.5) | 47 (30.7) | 14 (23.7) | 36 (26.7) | |
| Working prior | No | 41 (12.4) | 7 (8.1) | 7 (4.8) | 6 (11.3) | 25 (19.7) | 0.003 |
| | Yes | 290 (87.6) | 79 (91.9) | 139 (95.2) | 47 (88.7) | 102 (80.3) | |
| Occupation skill level | Professionals/Associate professionals | 86 (31.9) | 23 (31.5) | 50 (37.6) | 6 (13.3) | 26 (28.3) | 0.047 |
| | Trade, advanced clerical to elementary | 184 (68.1) | 50 (68.5) | 83 (62.4) | 39 (86.7) | 66 (71.7) | |
| CCI weighted condition | None | 288 (82.1) | 76 (86.4) | 131 (85.6) | 45 (76.3) | 106 (78.5) | 0.29 |
| | 1 or more | 63 (17.9) | 12 (13.6) | 22 (14.4) | 14 (23.7) | 29 (21.5) | |
| Pre-injury substance use condition | No | 321 (91.7) | 88 (100.0) | 149 (97.4) | 56 (94.9) | 129 (96.3) | 0.007 |
| | Yes | 29 (8.3) | <5 | <5 | <5 | 5 (3.7) | |
| Pre-injury mental health condition | No | 328 (93.7) | 85 (96.6) | 146 (95.4) | 56 (94.9) | 125 (93.3) | 0.78 |
| | Yes | 22 (6.3) | <5 | 7 (4.6) | <5 | 9 (6.7) | |
| Pre-injury disability | None | 284 (91.3) | 74 (89.2) | 125 (90.6) | 43 (93.5) | 96 (81.4) | 0.036 |
| | Mild to severe | 27 (8.7) | 9 (10.8) | 13 (9.4) | <5 | 22 (18.6) | |
| Place of injury | Road, street or highway | 276 (78.6) | 54 (61.4) | 148 (96.7) | 56 (94.9) | 109 (80.7) | <0.001 |
| | Other | 75 (21.4) | 34 (38.6) | 5 (3.3) | <5 | 26 (19.3) | |
| Injury severity (ISS), tertiles | < = 10 | 145 (41.3) | 35 (39.8) | 63 (41.2) | 21 (35.6) | 46 (34.1) | 0.55 |
| | 11–17 | 121 (34.5) | 36 (40.9) | 60 (39.2) | 20 (33.9) | 57 (42.2) | |
| | > = 18 | 85 (24.2) | 17 (19.3) | 30 (19.6) | 18 (30.5) | 32 (23.7) | |
| Nature of injury | Orthopaedic injuries | 133 (37.9) | 33 (37.5) | 66 (43.1) | 17 (28.8) | 48 (35.6) | 0.62 |
| | Head or SCI | 25 (7.1) | 8 (9.1) | 13 (8.5) | 6 (10.2) | 15 (11.1) | |
| | Chest, abdominal or multi trauma injuries | 193 (55.0) | 47 (53.4) | 74 (48.4) | 36 (61.0) | 72 (53.3) | |
| Number of claimants | Single claimant | 331 (94.3) | 83 (94.3) | 144 (94.1) | 54 (91.5) | 123 (91.1) | 0.70 |
| | Multiple claimants | 20 (5.7) | 5 (5.7) | 9 (5.9) | 5 (8.5) | 12 (8.9) | |
| Number of vehicles | One | 176 (50.1) | 72 (81.8) | 15 (9.8) | 18 (30.5) | 31 (23.0) | <0.001 |
| | Two or more | 175 (49.9) | 16 (18.2) | 138 (90.2) | 41 (69.5) | 104 (77.0) | |

*Notes*: All comparisons tested with Chi Square test, except for age (Kruskal-Wallis).

*Abbreviations*: CCI = Charlson Comorbidity Index; IRSAD = Index of Relative Socioeconomic and Disadvantage; ISS = Injury Severity Score; Q = quartile.

**Table 4. Characteristics of pedal cyclists stratified by fault attribution, n(%), N = 196.**

| | | No other at fault (n = 18) | Another at fault (n = 57) | Unknown (n = 121) | p-value |
|---|---|---|---|---|---|
| Sex | Male | 16 (88.9) | 43 (75.4) | 95 (78.5) | 0.48 |
| | Female | 2 (11.1) | 14 (24.6) | 26 (21.5) | |
| Age group (years) | Median [Q1, Q3] | 43 [27, 55] | 44 [35, 51] | 45 [34, 56] | 0.87 |
| University education | No | 10 (55.6) | 27 (47.4) | 65 (53.7) | 0.70 |
| | Yes | 8 (44.4) | 30 (52.6) | 56 (46.3) | |
| IRSAD, quintiles | 1, most disadvantaged | 1 (5.6) | 2 (3.6) | 9 (7.5) | 0.89 |
| | 2 | 2 (11.1) | 4 (7.1) | 10 (8.3) | |
| | 3 | 3 (16.7) | 11 (19.6) | 18 (15.0) | |
| | 4 | 6 (33.3) | 12 (21.4) | 34 (28.3) | |
| | 5, least disadvantaged | 6 (33.3) | 27 (48.2) | 49 (40.8) | |
| Preferred language English | Yes | 12 (66.7) | 43 (75.4) | 79 (65.3) | 0.39 |
| | No | 6 (33.3) | 14 (24.6) | 42 (34.7) | |
| Working prior | No | 2 (12.5) | 5 (9.1) | 19 (17.0) | 0.38 |
| | Yes | 14 (87.5) | 50 (90.9) | 93 (83.0) | |
| Occupation skill level | Professionals/Associate professionals | 7 (58.0) | 34 (71.0) | 56 (67.0) | 0.70 |
| | Trade, advanced clerical to elementary | 5 (42.0) | 14 (29.0) | 27 (33.0) | |
| CCI weighted condition | None | 15 (83.3) | 47 (82.5) | 102 (84.3) | 0.95 |
| | 1 or more | 3 (16.7) | 10 (17.5) | 19 (15.7) | |
| Place of injury | Road, street or highway | 18 | 57 | 120 | 0.73 |
| | Other | 0 | 0 | <5 | |
| Injury severity (ISS), tertiles | < = 10 | 7 (38.9) | 25 (43.9) | 55 (45.5) | 0.86 |
| | 11–17 | 6 (33.3) | 20 (35.1) | 34 (28.1) | |
| | > = 18 | 5 (27.8) | 12 (21.1) | 32 (26.4) | |
| Nature of injury | Orthopaedic injuries | 6 (33.3) | 24 (42.1) | 48 (39.7) | 0.95 |
| | Head or SCI | 4 (22.2) | 10 (17.5) | 20 (16.5) | |
| | Chest, abdominal or multi trauma injuries | 8 (44.4) | 23 (40.4) | 53 (43.8) | |

*Notes*: All comparisons tested with Chi Square test, except for age (Kruskal-Wallis).

*Abbreviations*: CCI = Charlson Comorbidity Index; IRSAD = Index of Relative Socioeconomic and Disadvantage; ISS = Injury Severity Score; Q = quartile.

control after driving on gravel or earth matter, or when travelling around a bend or corner. For the 68 motorcyclists who lost control on gravel or earth matter, 42 were driving on a road, street or highway, only two of whom reported that they were driving on a road with dirt or gravel surface in the injury event description. Other circumstances with no other at fault included motorcyclists who had a mechanical failure; had poor visibility, road or weather conditions (88.2% of which involved wet weather); slipped on tram tracks; or had a collision with or when attempting to avoid an animal, most of which involved kangaroos (n = 24, 68.6%). Three hundred (39.7%) motorcyclist injuries involved the motorcyclist losing control, of which 225 (72.1%) motorcyclists reported no other was at fault. Mechanical failures for motorcyclists predominantly included the brakes or wheels locking up or failing, stalling the motorcycle, or other mechanical factors (e.g., accelerator got stuck). Collision classifications in which there was predominantly no other at fault also included events in which the motorcyclist was injured while executing a jump (e.g., at a motor-cross track); or if the motorcyclist hit a curb or gutter, fence or wall, railing or barrier, pole or street sign, tree, or stationary vehicle. Claimants injured in collisions occurring at an intersection or roundabout predominantly reported that no other was at fault (51.7%), with 14 (48.3%) of those events occurring at a roundabout and only 9 (9.4%) of which referred to a turning vehicle.

**Table 5. Characteristics of pedestrians stratified by fault attribution, n(%), N = 354.**

| | | No other at fault (n = 79) | Another at fault (n = 100) | Unknown (n = 175) | p-value |
|---|---|---|---|---|---|
| Sex | Male | 51 (64.6) | 50 (50.0) | 88 (50.3) | 0.078 |
| | Female | 28 (35.4) | 50 (50.0) | 87 (49.7) | |
| Age group (years) | Median [Q1, Q3] | 48 [30, 74] | 48 [28, 66] | 61 [32, 76] | 0.014 |
| Education level | University | 8 (10.4) | 23 (24.5) | 33 (20.4) | 0.25 |
| | Completed high school | 10 (13.0) | 11 (11.7) | 22 (13.6) | |
| | Advanced diploma | 15 (19.5) | 22 (23.4) | 35 (21.6) | |
| | Did not complete high school | 44 (57.1) | 38 (40.4) | 72 (44.4) | |
| Region of residence | Regional and remote | 14 (18.7) | 7 (7.2) | 19 (10.9) | 0.062 |
| | Major cities | 61 (81.3) | 90 (92.8) | 155 (89.1) | |
| IRSAD, quintiles | 1, most disadvantaged | 14 (18.7) | 24 (24.7) | 29 (16.7) | 0.45 |
| | 2 | 8 (10.7) | 15 (15.5) | 26 (14.9) | |
| | 3 | 14 (18.7) | 9 (9.3) | 33 (19.0) | |
| | 4 | 14 (18.7) | 18 (18.6) | 37 (21.3) | |
| | 5, least disadvantaged | 25 (33.3) | 31 (32.0) | 49 (28.2) | |
| Preferred language English | Yes | 49 (62.0) | 58 (58.0) | 103 (58.9) | 0.85 |
| | No | 30 (38.0) | 42 (42.0) | 72 (41.1) | |
| Working prior | No | 38 (51.4) | 37 (41.1) | 108 (68.4) | <0.001 |
| | Yes | 36 (48.6) | 53 (58.9) | 50 (31.6) | |
| Occupation skill level | Professionals/Associate professionals | 11 (41) | 22 (49) | 18 (53) | 0.63 |
| | Trade, advanced clerical to elementary | 16 (59) | 23 (51) | 16 (47) | |
| CCI weighted condition | None | 46 (58.2) | 79 (79.0) | 110 (62.9) | 0.005 |
| | 1 or more | 33 (41.8) | 21 (21.0) | 65 (37.1) | |
| Pre-injury substance use condition | No | 68 (86.1) | 91 (92.9) | 151 (86.3) | 0.23 |
| | Yes | 11 (13.9) | 7 (7.1) | 24 (13.7) | |
| Pre-injury mental health condition | No | 72 (91.1) | 92 (93.9) | 151 (86.3) | 0.13 |
| | Yes | 7 (8.9) | 6 (6.1) | 24 (13.7) | |
| Pre-injury disability | None | 46 (75.4) | 65 (83.3) | 98 (70.5) | 0.11 |
| | Mild to severe | 15 (24.6) | 13 (16.7) | 41 (29.5) | |
| Place of injury | Road, street or highway | 65 (82.3) | 93 (93.0) | 159 (90.9) | 0.048 |
| | Other | 14 (17.7) | 7(7.0) | 16 (9.1) | |
| Injury severity (ISS), tertiles | < = 10 | 26 (32.9) | 41 (41.0) | 65 (37.1) | 0.75 |
| | 11–17 | 28 (35.4) | 30 (30.0) | 63 (36.0) | |
| | > = 18 | 25 (31.6) | 29 (29.0) | 47 (26.9) | |
| Nature of injury | Orthopaedic injuries | 26 (32.9) | 35 (35.0) | 65 (37.1) | 0.95 |
| | Head or SCI | 23 (29.1) | 27 (27.0) | 43 (24.6) | |
| | Chest, abdominal or multi trauma injuries | 30 (38.0) | 38 (38.0) | 67 (38.3) | |

*Notes*: All comparisons tested with Chi Square test, except for age (Kruskal-Wallis).

*Abbreviations*: CCI = Charlson Comorbidity Index; IRSAD = Index of Relative Socioeconomic and Disadvantage; ISS = Injury Severity Score; Q = quartile.

Collision classifications where there was predominantly **another at fault**, or the motorcyclist **claimed that another** was at fault, included collisions with: a turning vehicle or while turning; a vehicle turning from an adjacent street or driveway; a vehicle in or exiting a car park; a vehicle that was merging or changing lanes; or an oncoming vehicle. It should be noted that the collisions that involved a turning vehicle could have resulted in injury to the motorcyclist because they collided with another vehicle, or they lost control avoiding that vehicle and

**Table 6. Summary of the text used to describe the injury events.**

|  | N | Total words | Injury event description Median [Q1, Q3] Number of words per case |
|---|---|---|---|
| Motor vehicle drivers | 1150 | 14,599 | 9 [6, 16] |
| Motor cycle drivers | 786 | 13,021 | 13 [8, 21] |
| Pedal cyclists | 196 | 3042 | 11 [8, 18] |
| Pedestrians | 365 | 5007 | 10 [6, 18] |

*Abbreviations*: N = number, Q = Quartile.

fell from the motorcycle. Most types of injury events included less than a quarter of cases with **unknown fault**.

The remaining collision classifications not shown in Fig 5 included events where the claimant described falling off their bike but provided no other detail to allow for more precise classification of the injury event (n = 38) the majority of whom involved no other party at fault or the claimant denied that another was at fault (n = 29, 76.4%). Fewer than five claimants referred to a driver running a red light.

### Pedal cyclist injury event classifications

There were 24 cyclist injury event classifications, and only 10 classifications that included five or more claimants, Fig 6. While fault status was **unknown** for the majority of cases (median proportion of cases with unknown fault = 58.6%; range: 47.7–83.3%), collision classifications in which large proportions of cases recorded unknown fault status included events where the cyclist collided with the door of a parked vehicle, when the cyclist was hit in the rear by a vehicle travelling in the same direction, or when the cyclist hit a stationary or slowing vehicle in front of them. Less than 20% of all collision classifications involved no other at fault; however, collisions in which some cyclists were injured with **no other at fault** or who **denied that another was at fault**, involved a cyclist being hit by a turning or oncoming vehicle, while the cyclist was turning, or occurred at an intersection; or when a vehicle was stationary or slowing down in front of them (e.g., the vehicle was attempting to enter a car park). Injury event classifications in which there was predominantly **another at fault**, or the cyclist **claimed that**

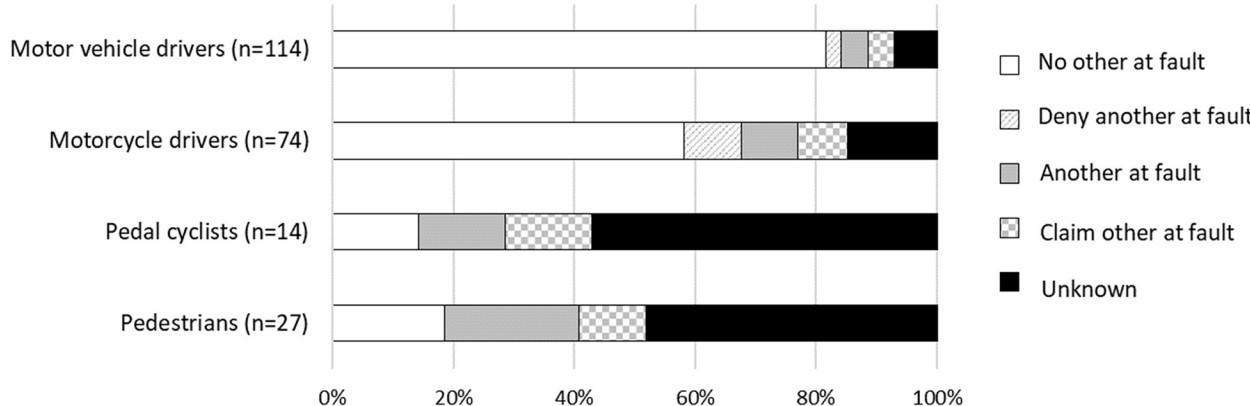

**Fig 2. Proportion of cases with no recollection of the injury event stratified by fault attributions by road user group, based on the terms used in the text description.**

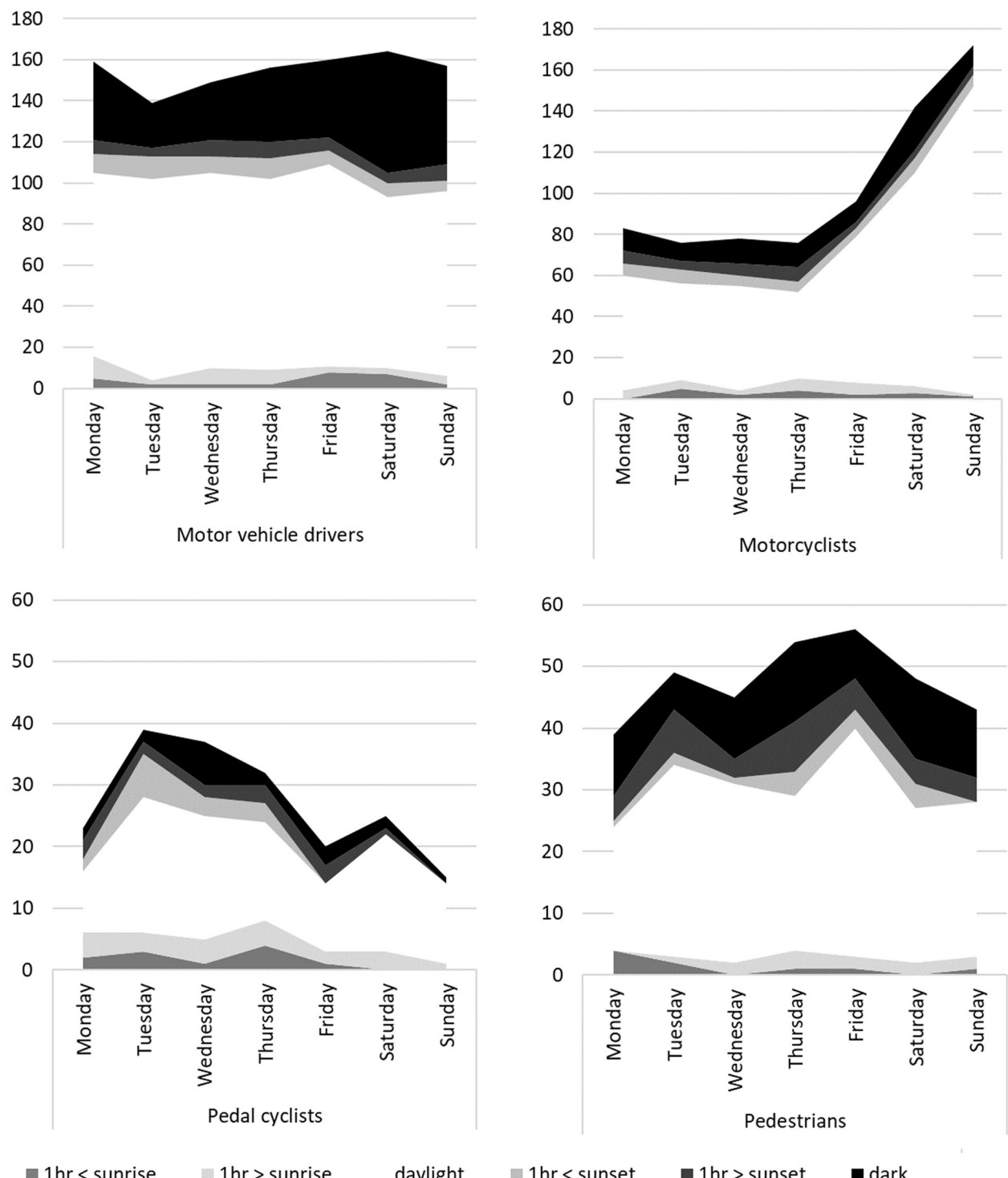

**Fig 3. The total number of injury events across different times of the day over the days of week for each road user group.**

**another was at fault**, occurred when there was a head-on collision; when a vehicle drove into the path of the cyclist from an adjacent direction (e.g., side street) or from another lane of traffic; when travelling through a roundabout or intersection; and when the event involved a head-on collision, or while attempting to avoid another vehicle, cyclists or object.

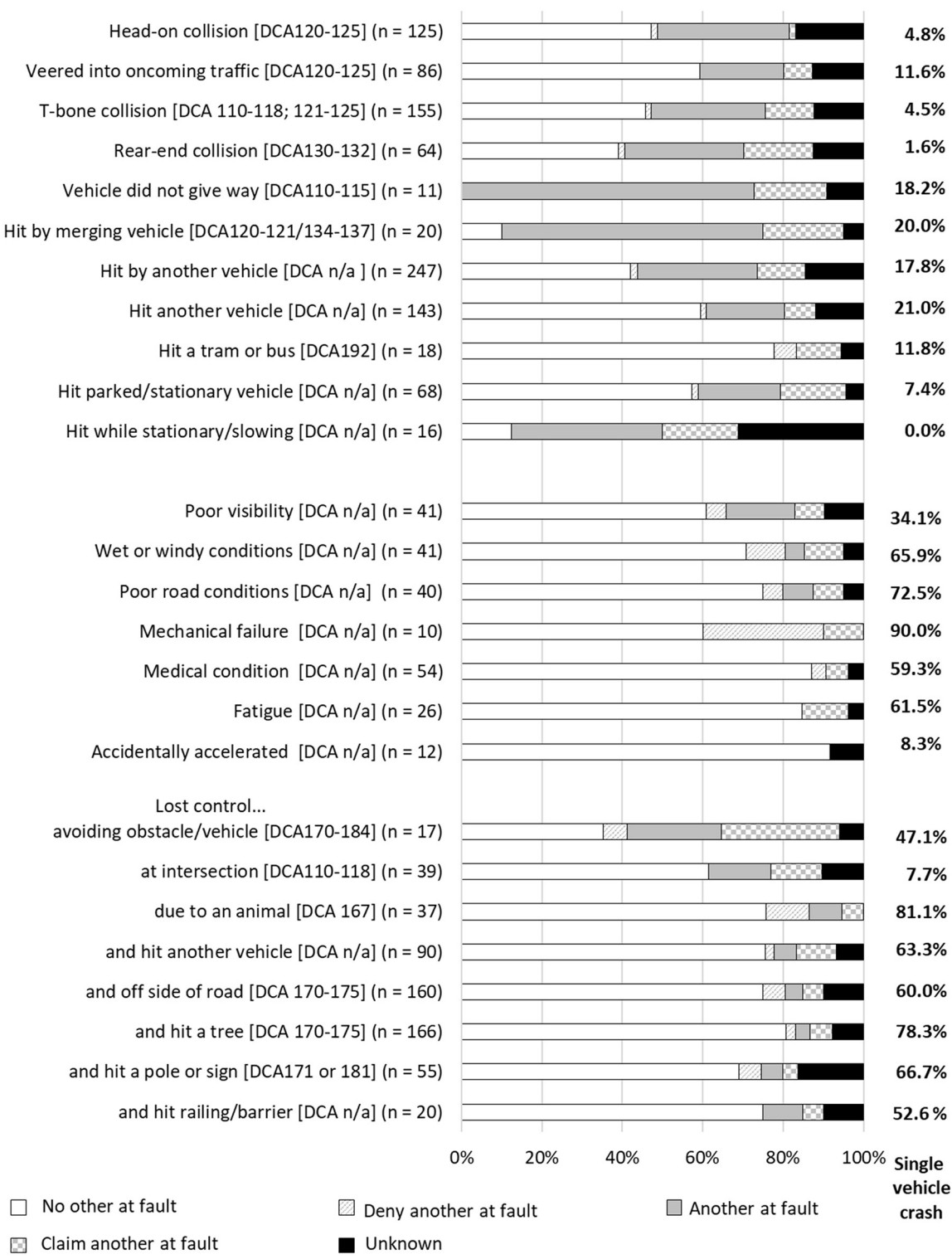

**Fig 4. Proportion of cases with each type of collision belonging to each fault group, drivers.**

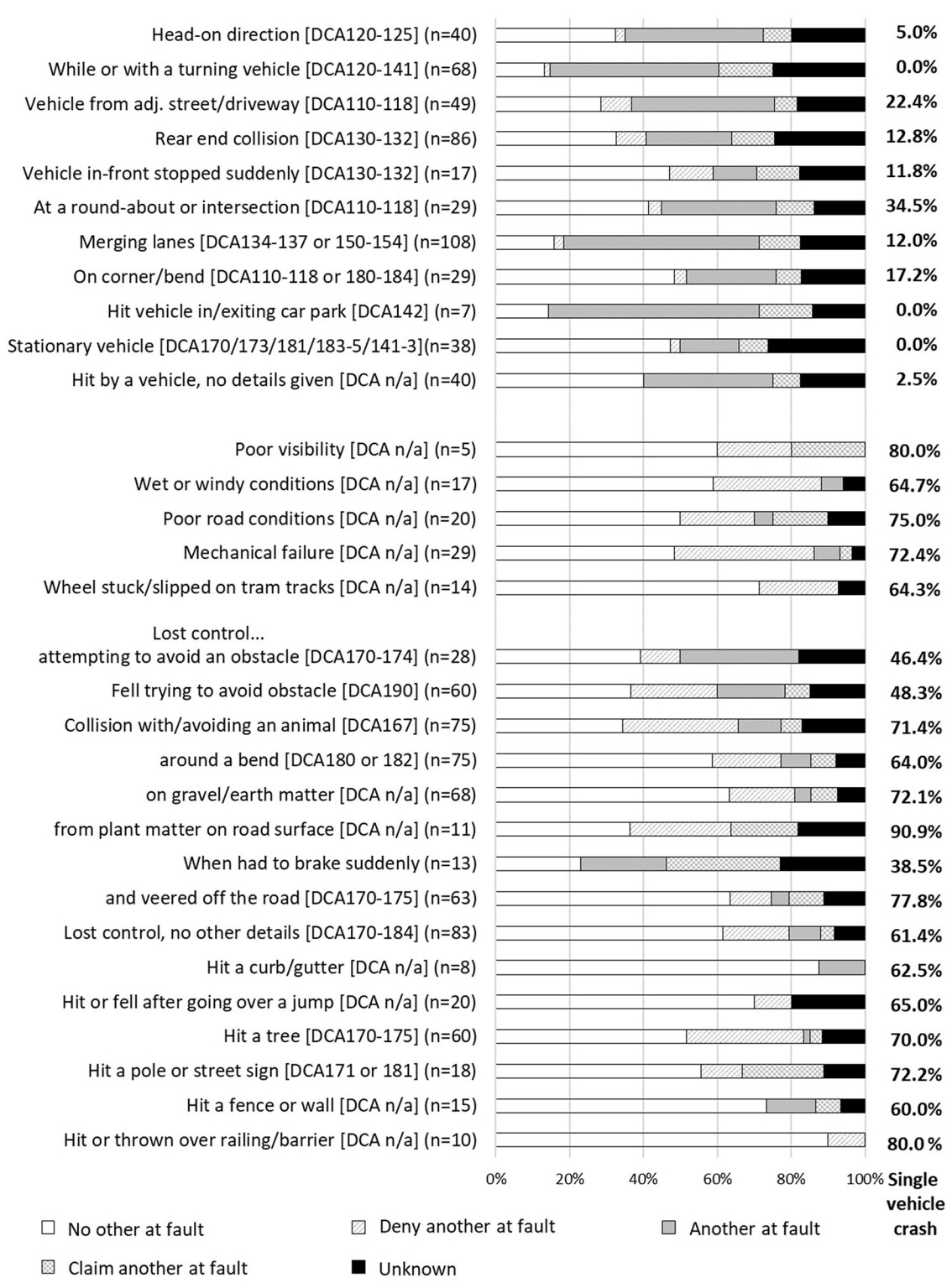

**Fig 5. Proportion of cases with type of collision belonging to each fault group, motorcyclists.**

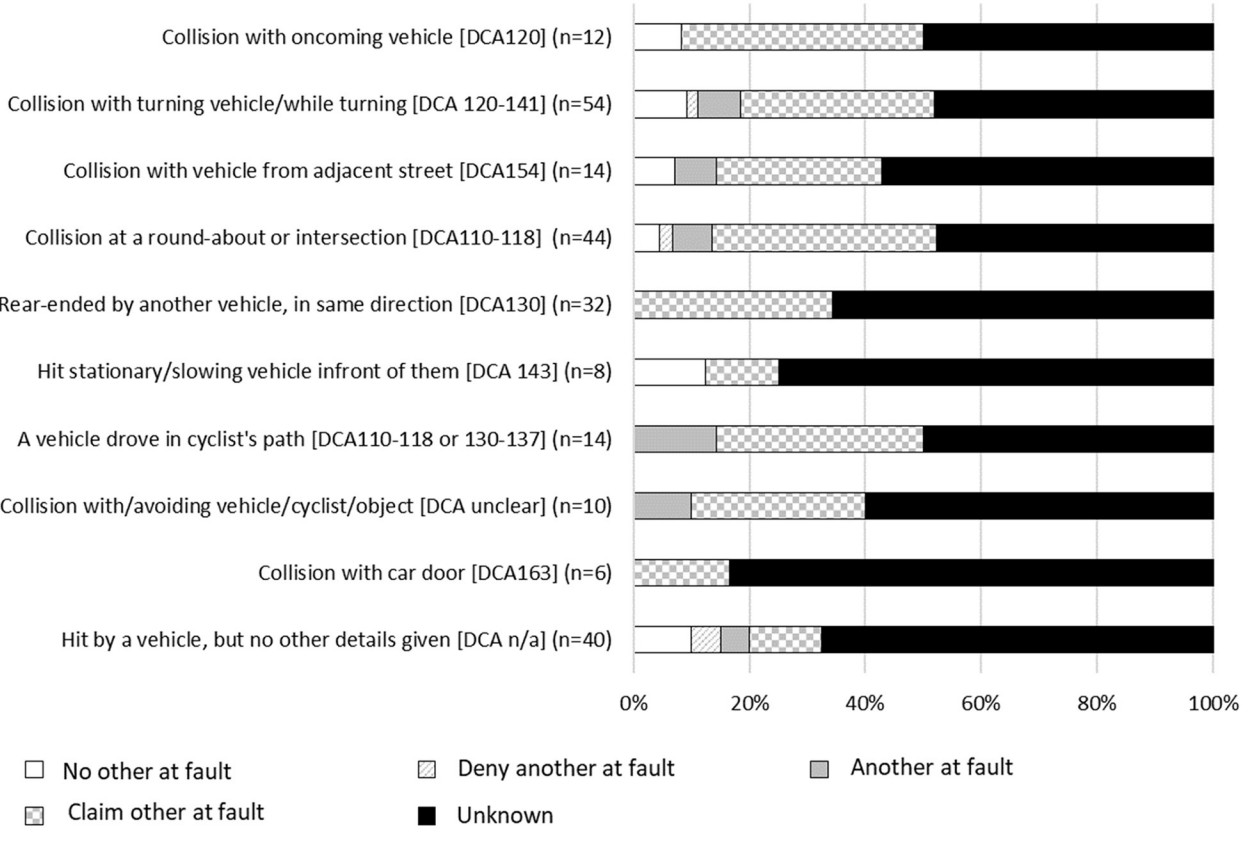

**Fig 6. Proportion of cases with each type of collision belonging to each fault group, pedal cyclists.**

Collision classifications that had fewer than five cases included events in which the cyclist was hit by a reversing vehicle or vehicle exiting a driveway; when riding or walking across the road at a pedestrian crossing; when the cyclist suddenly braked to avoid an obstacle or vehicle; or when they hit a vehicle that braked suddenly in front of them. Other collision classifications with few cases included events in which the cyclist or motor vehicle were manoeuvring with a U-turn or hook turn; while a vehicle was exiting a carpark into traffic; when the cyclist veered off the road, or slipped on tram or train tracks; when a motor vehicle ran a red light; or when there were poor weather conditions.

## Pedestrian injury event classifications

There were 13 injury event classifications in which pedestrians sustained injuries, 10 of which comprised five or more cases, Fig 7. Injury events for which a larger proportion of cases included **no other at fault**, or the pedestrian **denied that another was at fault**, included circumstances when the pedestrian was hit by a vehicle exiting a driveway; or in a "slow moving" traffic area (e.g., walking through a carpark, behind a vehicle reversing into a carpark, at a petrol station). The injury classifications in which a larger proportion of cases reported that **another was at fault**, or the pedestrian **claimed that another was at fault**, included circumstances when the pedestrian was exiting or entering their own vehicle; the pedestrian was hit while they were on a nature strip, median strip, footpath, or outdoor dining area by the side of a road; the pedestrian was hit while at the side of the road (e.g., in emergency lane, changing a tyre, or waiting to cross), or when a vehicle ran a red light. Injury event classifications with a

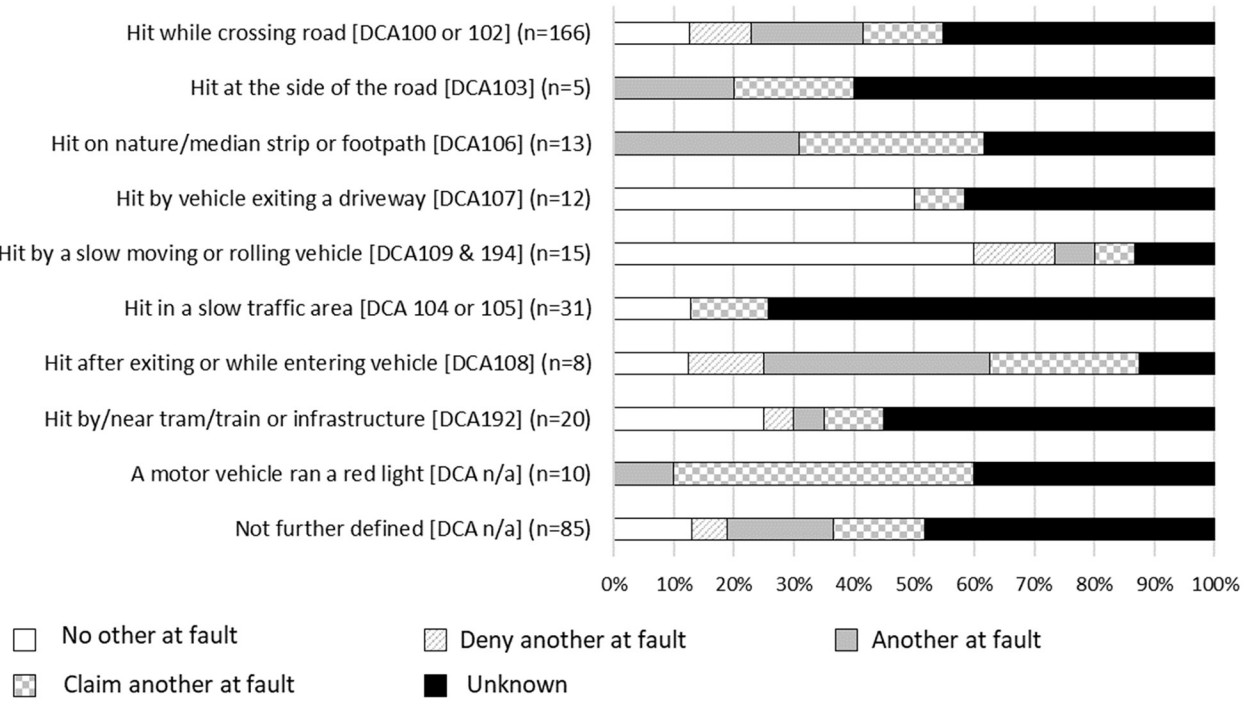

**Fig 7. Proportion of cases with each type of collision belonging to each fault group, pedestrians.**

high proportion of cases with **unknown fault** included a collision with the pedestrian in a slow traffic area; an impact with a tram or train, or infrastructure around a train or tram tracks; or when a pedestrian was hit at the side of the road. Injury event classifications that had fewer than five cases involved the pedestrian falling onto the road, being injured in a hit and run or through a deliberate act. All cases that recorded that the collision involved a "hit and run" reported unknown fault status, probably because the claimant cannot recall the event and it was not possible for the police to ascertain whether the other vehicle was at fault because they had left the scene.

## Discussion

Understanding the circumstances of road trauma events resulting in serious injury, particularly in relation to which vehicle was at fault, is important for advancing road safety and injury prevention strategies. In the present study of compensation claimants who survived a serious injury in Victoria, Australia, between 2010–2016, we examined crash circumstances using text mining, and evaluated the relationship between demographic, health, injury and injury characteristics with fault attributions. Two thirds of motor vehicle drivers and just over half of motorcyclists were injured in collisions where no other vehicle was at fault, or where the claimant denied that another was at fault, and more than a third of those injury events involved a single vehicle. For pedal cyclists and pedestrians, however, fault status was unknown for more than half of all cases. The vast majority of all road traffic injuries occurred during daylight hours, or within one hour of sunset, consistent with the fact that 77% of Victorian road use occurs between 7am and 7pm [29]. While there were some common patterns in the relationships between collision circumstances and fault attributions across road user groups, several features were specific to each road user group or type of road traffic collision. We now discuss the implications of the present findings for road safety and injury prevention.

## Implications for road safety

Understanding the circumstances leading to road traffic injury is crucial to improving road safety, and the implementation of countermeasures to reduce both the incidence and severity of road trauma [13, 30]. Most road traffic injuries are thought to occur due to human error [5, 31], even when environmental conditions or engineering and infrastructure factors are recognised to have played a role. Therefore, road safety strategies must consider how to reduce injury risk due to human error through all aspects of the Safe System approach, from improving road and infrastructure conditions to engineering or behavioural interventions and post-crash care to reduce both the incidence and impacts of human error.

Not surprisingly, road traffic injuries were common in circumstances where there are heightened opportunities for conflicts between road users, including being hit by another vehicle changing lanes or emerging from an adjacent direction, or approaching intersections or roundabouts. In many of these instances a large proportion of cases were injured at the fault of another vehicle, or the injured person claimed that another was at fault, perhaps due to the perception of human error or recklessness in the other driver. For instance, these injury events often involve driver distraction, driving in a manner that does not allow a safe braking distance, or misjudging the location of vehicles in other lanes [32]. Previous studies have shown that motor vehicle collision frequency is increased on roads with multiple shared or multiple lanes [33, 34] and at or approaching intersections [35]. In particular, intersection collisions often involve vehicles driving too closely behind another vehicle, turning or crossing in front of oncoming traffic, or violating traffic signals [32, 35, 36]. For cyclists, one of the most common on road collision events occurs when vehicles merge into or across the path of a cyclist [12], and the motor vehicle driver is typically at fault [37]. "Dooring" injuries were found to be infrequent for cyclists in the present cohort, which is consistent with previous studies in Australia [12, 38]. However, we recognise that cyclists may avoid dooring injuries by riding closer to traffic on roads with parked cars than they do on roads without parked cars [39]. Road safety may be enhanced with traffic calming strategies to reduce traffic volume and speed near intersections and collision hotspots for road users faced with turning or merging traffic, especially cyclists [34]. Moreover, protected infrastructure for pedal cyclists that separates them from both parked cars and traffic may reduce both the frequency and severity of cyclist injuries.

Pedestrians were predominantly older than the other road user groups, and were most often injured when crossing the road. We know that older pedestrians are particularly vulnerable to serious injury when crossing the road [7], possibly because it typically takes older adults more time to safely reach the other side of the road, particularly older people with comorbid conditions or frailty. Nature strips, traffic islands and median strips are common traffic calming measures that are used to promote pedestrian safety, and may provide a place of refuge for pedestrians who can cross the road in stages [6]. However, we found that five percent of pedestrians reported that they were injured on a median strip, footpath or outdoor dining area by the side of a road. In the absence of scene analysis, we speculate that most of those injury events probably occurred due to driver error rather than reckless pedestrian behaviour in which the driver breached the kerb or median to impact with the pedestrian. Moreover, while the specific road features of the injury locations was not known, previous studies have shown that the protective effects of median strips is lost if the median is not raised above the road surface, or if it is less than 150cm wide [34]. An analysis of the locations where pedestrians are more often injured near or on median strips should allow the identification of hotspot areas requiring enhanced safety strategies.

Nearly half of injured motor vehicle drivers, and one in ten motorcyclists, reported losing control in the collision. In most of those cases, there were no other vehicles in the collision and

no other vehicle was reported to be at fault. Rather, most people described situations where they lost traction when the vehicle veered into the gravel shoulder of the road, or into an embankment, or they veered off the road and hit another obstacle or rolled their vehicle. While pedal cyclist injuries have been found to often occur after the cyclist loses control in Canadian [40] and Australian studies [11, 41], these descriptors did not emerge in the text mining of the present pedal cyclist injury event descriptions. For motorised vehicles, previous research has found that single vehicle road traffic injuries often occur on roads with low shoulder width and sight distance where, presumably, the risk of losing traction or control of the vehicle is heightened if the driver veers onto the roadside shoulder [14]. We could not find any previous studies that reported the prevalence or circumstances of "loss of control" collisions for motorised vehicles. Most modern vehicles include safety features to detect when the driver is at risk of losing control (e.g., due to loss of traction or oversteering); however, these safety mechanisms are lacking for many people in the community given that 43% of vehicles registered in Australia are more than 10 years old [42] and electronic stability control systems only became mandatory for new vehicle registrations in Victoria since 2011 [43]. Given that many of the loss of control injury events occurred when drivers veered off road, or when motorcyclists were travelling around a bend or corner other environmental (e.g., rumble strips, roadside barriers, or other traffic calming interventions) and behaviour targets (e.g., safety campaigns via billboards in high risk zones, or broadcast via television, radio or social media) may have the greatest effect at reducing the risk of drivers losing control of their vehicle.

Many motor vehicle and motorcyclist collisions involved interactions with unsafe road conditions, losing control at the roadside shoulder or due to debris on the road, and collisions with trees, fences, poles or barriers at the roadside. One key target, therefore, to promote safety could be to improve the level of protection from fixed roadside objects with installation of roadside barriers, reductions in speed limits, or extension to the width of the sealed shoulder of rural roads [9]. That said, it is likely that the presence of roadside barriers reduced the severity of injuries sustained in those collisions (e.g., by preventing the injured person from colliding with oncoming traffic), and it is important that we do not assume that the presence of roadside barriers caused those collisions.

People often lost control in injury events in which they attempted to avoid a hazard or animal, especially kangaroos. In Australia, kangaroos are the most prevalent animal counterpart in road traffic collisions [30, 44], and the incidence of animal-vehicle collisions has increased since 2017 in Victoria [30]. The implementation of countermeasures is therefore a critical priority to reduce the burden from these types of injury events. Given that most animal-related injuries involved kangaroos, where injury risk is known to be heightened during the winter months and between dusk and midnight [44], injury prevention strategies should target high risk rural road networks and road users who are most likely to encounter kangaroos. Countermeasures may include imposing reduced speed limits between dusk and midnight, reviewing vehicle safety and driver skills, particularly for drivers who travel in high risk zones at high risk times of the day. Installation of road side barriers in high risk zones could also prevent drivers from veering into oncoming traffic or trees at the roadside while avoiding a kangaroo. While installation of under or overpasses to allow animals to cross roads can reduce risk in some settings [45], this is not feasible for Australia's extensive rural road network [44]. Moreover, those roadside barriers would need to account for the capacity of kangaroos to jump 2–3 metres in height. While driver education could reduce the incidence of animal-vehicle collisions, education-based strategies have not achieved these safety outcomes in other countries [46], and would require critical evaluation before large-scale investment in Australia.

Inclement weather conditions generally increase the risk of road traffic collisions, with no apparent difference in risk between moderate and heavy conditions [for a review, see: 13].

Weather conditions predominantly increase collision risk due to reductions in traction and visibility of hazards and other road users [47]. While only a small number of collisions cited a role of inclement conditions, we recognise that traffic volumes are typically reduced during periods of heavy fog, strong wind or rainfall. In particular, very few cyclists and only 20 motorcyclists referred to the role of inclement conditions in their collision, which is consistent with previous studies [12, 41], and highlights not only that fewer people probably choose to ride in inclement conditions but that those who do may be more experienced, fitter or cautious and therefore less likely to sustain an injury [48]. The prevention of road traffic injuries due to weather or visibility could be facilitated through targeted safety campaigns, broadcasting and traffic mitigation techniques during periods of poor weather, especially in winter or during heavy rainfall, and adjusting speed limits in hazard hotpots during inclement conditions [47]. Moreover, safety features in modern vehicles can reduce injury risk in all conditions, but particularly in wet conditions [32]. These features include anti-lock braking systems, lane departure warning, traction control, reversing camera and warning systems, as well as improved tyre quality and maintenance.

While we did not have information on the type or age of vehicles involved in the present collisions, a small number of motor vehicle and motorcyclist collisions involved mechanical failures highlighting the need for regular maintenance, particularly braking systems and tyre conditions given that these were the primary mechanical issues reported. Previous studies have shown that younger drivers may be particularly vulnerable to serious road traffic injury as they tend to drive older vehicles [10]. While few prior studies have examined the role of mechanical failures in road traffic injuries, a Canadian study published more than ten years ago also reported that mechanical failures were infrequent causes of serious road traffic injuries [49].

## Implications for understanding fault attribution

There are two major theories on how people attribute fault following a road traffic injury. First, the *actor-observer effect* is thought to generate biases in attributions of personal responsibility versus the responsibility of others. That is, people typically focus on the impact of situational and contextual factors (e.g., the weather, road conditions, unexpected hazards) on their own behaviour when they were at fault, but focus on intrinsic or dispositional characteristics of the other person when they are at fault [50]. Second, the *Defensive Attribution Theory* argues that when describing an injury event people tend to identify features of the event that were controllable and preventable, with a bias towards believing that another party was responsible for preventing the injury, particularly when the consequences are severe [51]. While it was not possible to quantitatively evaluate these biases in the present study, it does appear that people who did not attribute fault to another vehicle predominantly described the role of the environment or precipitating factors in the injury event, or their loss of control due to those factors. On the contrary, people who were injured when another was at fault typically described the other driver's contribution to the collision and the nature of the impact. It did not appear, however, that many people used emotive language or made statements about intrinsic characteristics of the other driver with the exception of a handful of cases (e.g., "... *was turning right with a green arrow ... an idiot driving an [imported model car] did not stop for her red light and she plowed [sic] into me*").

Few previous studies have examined which characteristics are associated with fault or responsibility attributions after injury. In one study, Gabbe, Simpson [52] reported characteristics associated with fault attribution in people who were hospitalised for orthopaedic road traffic injuries. Men were more often injured when no other was at fault or denied to be at

fault. A larger proportion of people injured when another was at fault, however, had a university level of education, were injured while cycling, and did not sustain a serious injury. Similarly, women and "not at fault" drivers were more likely to be injured in motor vehicle collisions in a study in Florida, USA, but were less likely to cause injuries to drivers of other vehicles [53]. Another recent study in Victoria, Australia, examined characteristics associated with being personally responsible for events resulting in serious injury [54]. People predominantly reported low levels of personal responsibility if they were female or if their injury was compensable. On the contrary, high levels of personal responsibility were associated with injuries from falls or motorcycle collisions compared with motor vehicle drivers, and for people with a pre-existing substance use condition. Unknown personal responsibility was associated with sustaining a head or spinal cord injury. Together with the present findings, therefore, it appears that people are more likely to be injured when another person is at fault if they are female and have higher socioeconomic position (i.e., higher levels of education, employment, or living in a neighbourhood with higher levels of socioeconomic advantage).

## Study strengths and limitations

This registry study has several strengths, particularly the inclusion of a very large sample of people who sustained serious injury after a road traffic collision. However, more than half of the potential participants could not be included as they did not have a text description of the injury event in their compensation claim, the majority of whom were injured in 2011–2012. In particular, the study disproportionately excluded eligible cases whose preferred language was English, who had a pre-existing substance use or mental health condition, sustained more severe injuries, were involved in injury events with fewer vehicles, or that occurred in the evening, and in collisions where another vehicle was at fault. The present results may therefore have under-represented the type or prevalence of different types of collisions in which another was at fault.

Even with relatively well structured text data, the processes involved in text mining is often described as part 'art' and part 'science' [55]. Immeasurable decisions are made when mining text data to refine and improve classifications through iterative data processing and dictionary development, and it is not possible to record and quantify every one of those steps and decisions made. In natural language processing or text mining research we typically seek to use coding rules and machine learning based processes, including the generation of robust dictionaries and coding rules using both a training set and a test set from the corpus, which may also be validated against an alternative coding system [56]. When preparing the dataset for text analysis we realised that it would not be wise to use natural language processing and machine learning techniques to extract meaning from the grammatical structure of the injury event description or to make assumptions about the actions of the various parties involved in an injury event given that it was often not possible for us to make these assumptions when we read the descriptions. Moreover, in several cases the injury event description only referred to the ambulance or police reports, which were not available to the researchers, and 229 people simply indicated that they could not recall the injury event in their text description. Many injury events were described in vague terms about the actions of their own vehicle or person versus other parties; e.g., in collisions that involved a turning vehicle it was not always clear whose vehicle was turning or the direction in which they were turning relative to the other vehicles involved. In some cases multiple claimants from the same injury event appeared to have the same text description. We assume that these cases probably had a linked claim for family members where individual claimants did not or could not provide a unique accident description. Therefore, we could not use the text to make assumptions about the individual claimant's personal attribution of fault. These are common problems in text descriptions of

injuries using claims data [55]. Unfortunately, the classification of injury events, such as the DCA coding [28], that are routinely generated by police and government transport authorities were not available to the study team, and so the injury event classifications that we generated could not be validated. Instead, we regularly cross checked the classifications against the full text description to ensure that the classifications were accurate.

Finally, given that the text data were taken from claims submitted to the compensation scheme, the text descriptions probably lacked details about potential negligent or reckless behaviour by the claimant during the injury event (e.g., use of a mobile phone, fatigue, or drug/alcohol use) because they were not asked about it when making their claim, because the claimant wanted to reduce the risk that they could be charged for dangerous driving causing serious injury or death, or to maximise their entitlements for additional benefits (e.g., loss of earnings) or common law damages. Moreover, the median length of the text descriptions was 11 words, which highlights that the majority of claimants did not provide detailed descriptions of the injury event.

Despite the present limitations, this study has generated novel insights and resources through the use of iterative text mining and processing. The comprehensive dictionaries that we developed could be used by compensation schemes, or other researchers, who seek to understand the patterns and incidence of different types of collision classifications using routinely collected administrative or insurance data. We recommend now that future studies extend the methods used in the present study using predictive modelling to generate deeper insights into the association between fault attribution and/or traffic collision characteristics (e.g., single vehicle versus multiple vehicles) and circumstances (e.g., environmental conditions, driver characteristics) on a range of factors, including crash severity, injury severity, survival rates and long-term outcomes.

## Conclusions

The present study has presented a novel overview of compensable road traffic collision circumstances resulting in serious injury. Using text mining we characterised the predominant types of road traffic injury events that occur when another vehicle is at fault, or not, and have identified potential strategies to enhance road safety. Future research should now examine whether the implementation of countermeasures has played a role in reducing the incidence and severity of the types of collisions that have been targeted, from animal-related and loss of control collisions through to multi-vehicle, vehicle-cyclist, and vehicle-pedestrian conflicts.

## Supporting information

**S1 File.**
(DOCX)

**S2 File.**
(DOCX)

**S3 File.**
(DOCX)

**S4 File.**
(DOCX)

## Acknowledgments

We received advice on text mining procedures in QDA Miner from Mr David White, Survey Design and Analysis Services Pty Ltd, and acknowledge the contribution of Ms Georgina Lau

for obtaining the sunrise and sunset times from Geosciences Australia to classify the time of day of injury events. We also acknowledge the TAC road safety and business intelligence teams, which provided important guidance for the analysis and synthesis of the injury events.

## Author Contributions

**Conceptualization:** Melita J. Giummarra, Ben Beck, Belinda J. Gabbe.

**Data curation:** Melita J. Giummarra, Belinda J. Gabbe.

**Formal analysis:** Melita J. Giummarra.

**Funding acquisition:** Melita J. Giummarra.

**Investigation:** Melita J. Giummarra, Ben Beck, Belinda J. Gabbe.

**Methodology:** Melita J. Giummarra, Ben Beck, Belinda J. Gabbe.

**Project administration:** Melita J. Giummarra.

**Resources:** Melita J. Giummarra.

**Supervision:** Belinda J. Gabbe.

**Writing – original draft:** Melita J. Giummarra.

**Writing – review & editing:** Melita J. Giummarra, Ben Beck, Belinda J. Gabbe.

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
