## [Decision Letter · Decision Letter 0]

11 Dec 2020

PONE-D-20-34781

Classification of road traffic injury collision characteristics using text mining analysis: Implications for road injury prevention

PLOS ONE

Dear Dr. Giummarra,

Thank you for submitting your manuscript to PLOS ONE. After careful consideration, we feel that it has merit but does not fully meet PLOS ONE’s publication criteria as it currently stands. Therefore, we invite you to submit a revised version of the manuscript that addresses the points raised during the review process.

We look forward to receiving your revised manuscript.

Kind regards,

Feng Chen

Academic Editor

PLOS ONE

Journal Requirements:

3. We note you have included a table to which you do not refer in the text of your manuscript. Please ensure that you refer to Table 6 in your text; if accepted, production will need this reference to link the reader to the Table.

5. Please upload a copy of Supporting Materials 1,2, 3 which you refer to in your text on page 8.

Reviewers' comments:

Reviewer's Responses to Questions

**Comments to the Author**

1. Is the manuscript technically sound, and do the data support the conclusions?

Reviewer #1: Yes

Reviewer #2: Yes

2. Has the statistical analysis been performed appropriately and rigorously? 

Reviewer #1: No

Reviewer #2: Yes

3. Have the authors made all data underlying the findings in their manuscript fully available?

Reviewer #1: No

Reviewer #2: Yes

4. Is the manuscript presented in an intelligible fashion and written in standard English?

Reviewer #1: Yes

Reviewer #2: Yes

5. Review Comments to the Author

Reviewer #1: This paper investigates road traffic injury collision characteristics in Victoria, Australia. The investigation is based on the data from Victorian State Trauma Registry using text mining techniques, which provide many detailed and structured information on occupants involved in the reported crashes. The research topic is interesting and worth of investigation. While the occupant-specific factors are enriched significantly, developing statistical regression models (such as logit/probit model) would provide a deeper insight on the effects of these factors (as well as those specific to vehicle, roadway, and environment, etc.) on crash injury severity, which is consistent with many previous studies on traffic safety analysis. Given the length of this article, it can be a direction for future research.

Besides, driver at fault or not is a key factor considered in the empirical analysis. More existing findings on its safety effect can be referred to support the results in this research. For example, the following article analyzed the effects of driver at fault on the injury severities of the own and other driver-vehicle units in two-vehicle crashes:

The interactive effect on injury severity of driver-vehicle units in two-vehicle crashes. Journal of Safety Research, 2016, 59: 105-111.

Reviewer #2: This paper characterised crash characteristics of road traffic collisions in Victoria, Australia, and examined the relationship between crash characteristics and fault attribution. The topic is interesting. The methods sound. The results are meaningful and useful. It is suggested to be published.

6. PLOS authors have the option to publish the peer review history of their article (what does this mean?). If published, this will include your full peer review and any attached files.

Reviewer #1: No

Reviewer #2: No

---

## [Author Response · Author response to Decision Letter 0]

21 Dec 2020

Reviewer #1: This paper investigates road traffic injury collision characteristics in Victoria, Australia. The investigation is based on the data from Victorian State Trauma Registry using text mining techniques, which provide many detailed and structured information on occupants involved in the reported crashes. The research topic is interesting and worth of investigation. 

While the occupant-specific factors are enriched significantly, developing statistical regression models (such as logit/probit model) would provide a deeper insight on the effects of these factors (as well as those specific to vehicle, roadway, and environment, etc.) on crash injury severity, which is consistent with many previous studies on traffic safety analysis. Given the length of this article, it can be a direction for future research.

RESPONSE: This is a very good suggestion. In the initial draft of the paper we had included multinomial logistic regression to determine which characteristics were associated with being at fault for each of the road user groups. A team decision was made, however, to omit this from the paper as the data presented from the text mining and descriptive statistics were already quite complex and detailed. 

We have taken this comment on board in our revision, however, and have included a suggestion on Page 37 for future research to “extend the methods used in the present study using predictive modelling in order to generate deeper insights into the association between fault attribution and/or traffic collision characteristics (e.g., single vehicle versus multiple vehicles) and circumstances (e.g., environmental conditions, driver characteristics) on a range of factors, including crash severity, injury severity, survival rates and long-term outcomes.”

Besides, driver at fault or not is a key factor considered in the empirical analysis. 

RESPONSE: That is correct. Unfortunately the fault-related data recorded in the compensation claim lodgment only relates to whether another party was at fault. This is because people whose injuries occur when another is partially or fully at fault may be entitled to additional compensation through a common law claim. The compensation scheme does not typically need to know whether the injured person was also at fault. They only need to know whether they were reckless/negligent (e.g., due to intoxication) or intentionally caused the collision and were convicted of culpable offences, which would limit their entitlements. We did not have access to those data through the compensation claim records

More existing findings on its safety effect can be referred to support the results in this research. For example, the following article analyzed the effects of driver at fault on the injury severities of the own and other driver-vehicle units in two-vehicle crashes: The interactive effect on injury severity of driver-vehicle units in two-vehicle crashes. Journal of Safety Research, 2016, 59: 105-111.

RESPONSE: Thank you for this suggestion. We have now referred to that study on Page 35, to highlight the findings that were consistent with the two Australian studies examining fault attribution.

Reviewer #2. This paper characterised crash characteristics of road traffic collisions in Victoria, Australia, and examined the relationship between crash characteristics and fault attribution. The topic is interesting. The methods sound. The results are meaningful and useful. It is suggested to be published.

RESPONSE: We thank Reviewer 2 for their positive comments on our manuscript.

---

## [Decision Letter · Decision Letter 1]

5 Jan 2021

Classification of road traffic injury collision characteristics using text mining analysis: Implications for road injury prevention

PONE-D-20-34781R1

Dear Dr. Giummarra,

We’re pleased to inform you that your manuscript has been judged scientifically suitable for publication and will be formally accepted for publication once it meets all outstanding technical requirements.

Kind regards,

Feng Chen

Academic Editor

PLOS ONE

Additional Editor Comments (optional):

Reviewers' comments:

Reviewer's Responses to Questions

**Comments to the Author**

1. If the authors have adequately addressed your comments raised in a previous round of review and you feel that this manuscript is now acceptable for publication, you may indicate that here to bypass the “Comments to the Author” section, enter your conflict of interest statement in the “Confidential to Editor” section, and submit your "Accept" recommendation.

Reviewer #1: All comments have been addressed

Reviewer #2: (No Response)

2. Is the manuscript technically sound, and do the data support the conclusions?

Reviewer #1: (No Response)

Reviewer #2: (No Response)

3. Has the statistical analysis been performed appropriately and rigorously? 

Reviewer #1: (No Response)

Reviewer #2: (No Response)

4. Have the authors made all data underlying the findings in their manuscript fully available?

Reviewer #1: (No Response)

Reviewer #2: (No Response)

5. Is the manuscript presented in an intelligible fashion and written in standard English?

Reviewer #1: (No Response)

Reviewer #2: (No Response)

6. Review Comments to the Author

Reviewer #1: (No Response)

Reviewer #2: (No Response)

7. PLOS authors have the option to publish the peer review history of their article (what does this mean?). If published, this will include your full peer review and any attached files.

Reviewer #1: No

Reviewer #2: No

---

## [Editor Report · Acceptance letter]

18 Jan 2021

PONE-D-20-34781R1 

Classification of road traffic injury collision characteristics using text mining analysis: Implications for road injury prevention 

Dear Dr. Giummarra:

I'm pleased to inform you that your manuscript has been deemed suitable for publication in PLOS ONE. Congratulations! Your manuscript is now with our production department. 

Kind regards, 

on behalf of

Dr. Feng Chen 

Academic Editor

PLOS ONE